# FLOW OF SPANS: GENERALIZING LANGUAGE MODELS TO DYNAMIC SPAN-VOCABULARY VIA GFLOWNETS

**Bo Xue**[1,3*]**, Yunchong Song**[2*]**,**
**Fanghao Shao**[1]**, Xuekai Zhu**[1]**, Lin Chen**[1]**, Luoyi Fu**[3]**, Xinbing Wang**[3]**, Zhouhan Lin**[1,2,4†]

[1]LUMIA Lab, School of Artificial Intelligence, Shanghai Jiao Tong University
[2]Shanghai Artificial Intelligence Laboratory
[3]Shanghai Jiao Tong University    [4]Shanghai Innovation Institute

`sappho_x@sjtu.edu.cn, songyunchong@pjlab.org.cn,`
`lin.zhouhan@gmail.com`

## ABSTRACT

Standard autoregressive language models generate text token-by-token from a fixed vocabulary, inducing a *tree-structured state space* when viewing token sampling as an action, which limits flexibility and expressiveness. Recent work introduces dynamic vocabulary by sampling retrieved text spans but overlooks that the same sentence can be composed of spans of varying lengths, lacking explicit modeling of the *directed acyclic graph (DAG) state space*. This leads to restricted exploration of compositional paths and is biased toward the chosen path. Generative Flow Networks (GFlowNets) are powerful for efficient exploring and generalizing over state spaces, particularly those with a DAG structure. However, prior GFlowNets-based language models operate at the token level and remain confined to tree-structured spaces, limiting their potential. In this work, we propose **F**low **o**f **S**pan**S** (**FoSS**), a principled GFlowNets framework for span generation. FoSS constructs a dynamic span vocabulary by segmenting the retrieved text flexibly, ensuring a DAG-structured state space, which allows GFlowNets to explore diverse compositional paths and improve generalization. With specialized reward models, FoSS generates diverse, high-quality text. Empirically, FoSS improves MAUVE scores by up to 12.5% over Transformer on text generation and achieves 3.5% gains on knowledge-intensive tasks, consistently outperforming state-of-the-art methods. Scaling experiments further demonstrate FoSS benefits from larger models, more data, and richer retrieval corpora, retaining its advantage over strong baselines. Code is available at `https://github.com/sappho-x/Flow-of-Spans`.

## 1 INTRODUCTION

Standard autoregressive language models generate text token-by-token from a fixed, finite vocabulary (Brown et al., 2020; Radford et al., 2019; Sennrich et al., 2016). However, this approach has inherent limitations. The fixed vocabulary restricts the granularity of generation, and the resulting tree-structured state space, as illustrated in Figure 1, where each state has a unique prefix predecessor, limits the model's ability to efficiently explore alternative generation paths. In such a structure, each sample can only follow a single path, whereas a directed acyclic graph (DAG) structure allows simultaneous exploration of multiple paths, covering a broader region of the state space (see the green area in the DAG and tree state space in Figure 1). To address these limitations, recent work has introduced dynamic vocabularies and variable-length generation units, enabling models to construct context-aware, multi-word vocabularies (Cao et al., 2024; Martins et al., 2022; Li et al., 2024; Lan et al., 2023). However, these methods overlook the inherent DAG nature of sentence composition, where the same sentence (e.g., "ABCDEFGH") can be formed through different span combinations, such as "AB"→"CD"→"EFGH" or "AB"→"C"→"DE"→"FG"→"H". Consequently, existing ap-

---

*Equal contribution. † Corresponding author.

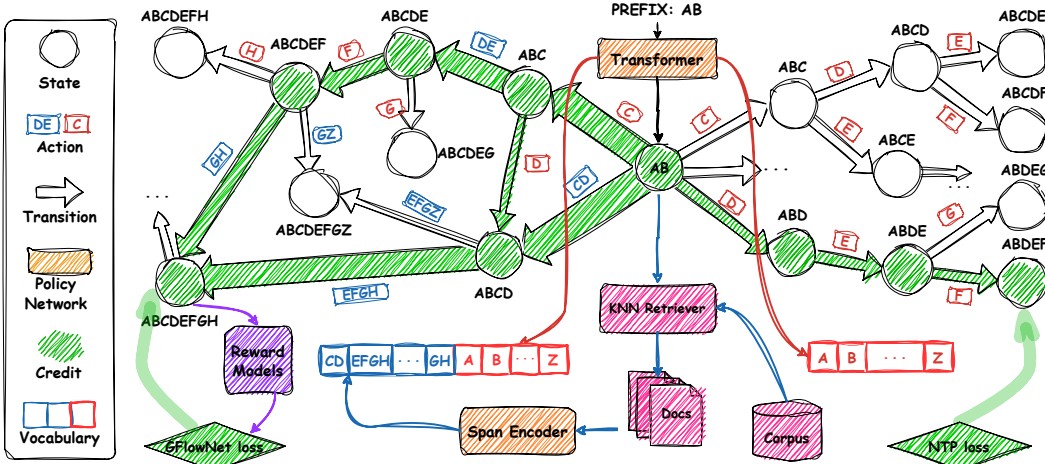

Figure 1: Given the prefix, a standard language model generates token-by-token and forms a *tree state space*, trained with next-token prediction (NTP) loss; while in FoSS, since a sentence can be composed of spans in multiple ways, we construct a *DAG state space* and optimize it with GFlowNets.

proaches do not explicitly model the DAG-structured state space, relying instead on training data to incidentally reflect such structures and lacking theoretical guarantees for coverage or generalization.

Generative Flow Networks (GFlowNets) are a promising method for solving the aforementioned problems. GFlowNets are a class of generative model that samples with a probability proportional to the reward, known for their excellent performance and theoretical guarantees in exploring and generalizing over state spaces, especially those with a DAG structure (Shen et al., 2023; Krichel et al., 2024; Atanackovic & Bengio, 2024). There has been limited work on applying GFlowNets to language models (Hu et al., 2024; Yu et al., 2025; Lee et al., 2025), and these efforts primarily treat GFlowNets as a trainable alternative sampling strategy for language models, without constructing a DAG-structured state space. This reduces the GFlowNets framework to a degraded learning paradigm that limits the exploration of state transitions and is unable to fully leverage the potential of GFlowNets (Li et al., 2023; Shen et al., 2023; Malkin et al., 2022; Bengio et al., 2023). In this work, we introduce the GFlowNets framework to span language models for the first time, explicitly constructing a DAG-structured state space that allows GFlowNets to fully unleash its power in the context of span generation. At the same time, unlike previous scenarios such as reasoning, where clear answers are available (Yu et al., 2025), we explore the design of reward models within the GFlowNets framework to enhance the performance of general text generation.

We propose FoSS, a span language model within a principled GFlowNets framework. FoSS introduces a flexible text segmentation algorithm to construct the DAG-structured state space. Then, we explicitly model the DAG space with GFlowNets, enabling efficient exploration and generalization over the state space. Benefiting from the specialized reward models, FoSS can be efficiently trained while maintaining both the quality and diversity of text generation. Our extensive evaluation demonstrates that FoSS significantly outperforms existing methods across various tasks. In text generation, FoSS achieves a 5.51% MAUVE score improvement over the state-of-the-art, while attaining consistently higher scores in GPT-4-based evaluations, which aligned with human preferences (Zheng et al., 2023). In the domain adaptation task, FoSS outperforms fully fine-tuned models without any training, showing robust generalization ability. Comprehensive ablation studies show that without the DAG-state space design or GFlowNets training, the model's performance drops sharply, validating our design as an indispensable component. Furthermore, scaling experiments confirm that FoSS effectively leverages larger model capacities, increased training data, and richer retrieval corpora, revealing the tremendous potential of FoSS when allocating a larger computational budget.

## 2 PRELIMINARIES

GFlowNets are designed to learn stochastic policies that sample from an unnormalized distribution over complex objects through a sequence of discrete actions within a Markov Decision Process (MDP) framework (Pan et al., 2024; Jain et al., 2022; Shen et al., 2023). Let the state space be $\mathcal{S}$

---

**Algorithm 1:** FoSS Training

---

**Input:** Training corpus $\mathcal{D}$, Forward policy $P_F(\cdot|\cdot; \theta)$, Reward function $R(\cdot)$ (Eq. 3), Learning
   rate $\eta$, Bernoulli mixing probability $\pi$
**Output:** Fine-tuned parameters $\theta^\star$

---

**Pre-processing:** Apply DAG-Inducing Span Segmentation algorithm to $\mathcal{D}$ to obtain $\mathcal{D}'$
Initialize prefix-trajectory buffer $\mathcal{B} \leftarrow \emptyset$, epoch counter $e \leftarrow 0$
**while** *not converged* **do**
 |   $e \leftarrow e + 1$          // Mini-batch implicitly handled
 |   **foreach** *segmented document* $d \in \mathcal{D}'$ **do**
 |  |   Truncate $d$ to prefix $c$ and residual $d \setminus c$
 |  |   Build vocabulary $V$ conditioned on $d$
 |  |   **if** $e = 1$ **then**
 |  |  |   $\tau \leftarrow$ construct trajectory using spans from $d \setminus c$ as action sequence
 |  |  |   $\mathcal{B} \leftarrow \mathcal{B} \cup \{(\tau, R(\tau))\}$
 |  |   **else**
 |  |  |   **if** $\mathrm{Bernoulli}(\pi) = 1$ **then**     // Online sampling
 |  |  |  |   $\tau \leftarrow$ sample trajectory from $P_F(\cdot|\cdot; V, \theta)$ from $s_0 = c$
 |  |  |  |   $\mathcal{B} \leftarrow \mathcal{B} \cup \{(\tau, R(\tau))\}$
 |  |  |   **else**          // offline sampling
 |  |  |  |   $\tau \leftarrow$ sample trajectory from $\mathcal{B}(c)$ and compute $P_F(\cdot|\cdot; V, \theta)$
 |   $\theta \leftarrow \theta - \eta \nabla_\theta \mathcal{L}(\tau; \theta)$ (Eq. 2)
**return** $\theta^\star \leftarrow \theta$

---

and the set of edges be $\mathbb{E} \subset \mathcal{S} \times \mathcal{S}$ for a directed acyclic graph (DAG) $G = (\mathcal{S}, \mathbb{E})$, where vertices represent states and edges represent transitions between states. The process begins from an initial state $s_0 \in \mathcal{S}$, with no incoming edges, and terminates at a terminal state $x \in \mathcal{X} \subseteq \mathcal{S}$, where $\mathcal{X}$ is the set of terminal states.

Each complete trajectory $\tau = (s_0 \to s_1 \to \cdots \to s_n = x)$ consists of a sequence of states, with transitions $(s_i \to s_{i+1}) \in \mathbb{E}$ corresponding to actions taken at each step. The objective of GFlowNets is to learn a forward policy $P_F(s_{i+1}|s_i)$, which defines the probability of transitioning from one state to the next. The probability of a trajectory $\tau$ is the product of the transition probabilities along the sequence: $P_F(\tau) = \prod_{i=0}^{n-1} P_F(s_{i+1}|s_i)$.

The terminal distribution $P_F^\top(x)$ represents the marginal distribution over terminal states, defined as: $P_F^\top(x) = \sum_{\tau \to x} P_F(\tau)$, where the sum is taken over all trajectories $\tau$ that terminate in terminal state $x$. The objective of GFlowNets is to learn the forward policy such that the terminal distribution $P_F^\top(x)$ is proportional to a reward function $R(x)$, i.e., $P_F^\top(x) \propto R(x)$, where $R(x)$ is an unnormalized density over terminal states (Bengio et al., 2023). To achieve this, GFlowNets associate a flow with each state in the graph. Concretely, a state-flow function $F : \mathcal{S} \to \mathbb{R}_+$ is defined that assigns a non-negative value to each state, representing the total flow passing through it. This function enables various training objectives for GFlowNets. In addition to the forward policy, GFlowNets incorporate a backward policy $P_B(s_i|s_{i+1})$, which sequentially deconstructs compositional objects, transforming the intractable matching of marginal distributions over terminal states into a tractable one of matching distributions over complete trajectories (Gritsaev et al., 2025; Jang et al., 2024).

## 3 SPAN-GENERATION WITH GFLOWNETS

In this section, we introduce our novel span-based text generation approach utilizing GFlowNets. Prior language modeling approaches typically operate with word-level vocabularies where each element is referred to as a token. These static vocabularies impose constraints on generation granularity. In our framework, vocabulary elements can be either single words or multi-word phrases, which we collectively refer to as spans. We first formalize the span-generation problem within a MDP framework, establishing a comprehensive formulation that enables flexible span sampling. Then, we

present our learning objective and training policy that effectively navigates the DAG-structured state space. Furthermore, we detail the key components that parameterize our model: the policy network architecture that generates spans from a dynamic vocabulary, and our reward function formulation that balances text fluency and alignment with human-written text.

## 3.1 Markov Decision Process Formulation for GFlowNets Learning

To formalize span-based text generation within the GFlowNets framework, we establish a complete MDP formulation with the following components: **State**: Each state $s_i \in \mathcal{S}$ is represented as a string $s_i = \text{CONCAT}(c, t_i)$, where $c$ is the prefix text, $t_i$ denotes the text generated so far consisting of previously sampled spans, and $\text{CONCAT}$ represents string concatenation. **Action**: We denote by $a$ a candidate action from the action space, and by $a_i$ the actual action sampled. An action $a_i$ involves selecting a span to be appended to the current state $s_i$. **Transition**: Given a state $s_i$ and an action $a_i$, the transition function deterministically yields the next state $s_{i+1}$ by appending the selected span to $s_i$. **Reward**: The reward function $R : \mathcal{X} \to \mathbb{R}^+$ assigns a non-negative value to each terminal state $x \in \mathcal{X} \subseteq \mathcal{S}$, where we employ specialized reward models to evaluate the quality and coherence of the generated text.

Since actions involve spans of variable length, multiple distinct trajectories can lead to the same state, ensuring the underlying graph $G$ is a DAG. For example, in Figure 1, the state "ABCD" can be reached via two different trajectories: (AB → ABCD) and (AB → ABC → ABCD). This contrasts with conventional token-by-token autoregressive language models, where each state can only be reached by a single unique trajectory, forming a tree (Lee et al., 2025; Hu et al., 2024; Yu et al., 2025). As shown in Figure 1, token-by-token autoregressive language models generate single words sequentially with accumulative prefixes. This structure restricts each non-initial state to have exactly one parent node. For example, the state "ABCD" can only be reached from state "ABC", creating a rigid tree structure. This makes the backward policy deterministic with $P_B(s_i|s_{i+1}) = 1$. In such degenerated case, GFlowNets reduces to discrete-action soft Q-learning (Malkin et al., 2022), limiting the number of possible transitions and impeding exploration of the expression space (Li et al., 2023).

We instantiate the forward policy $P_F$ with a span language model parameterized by $\theta$, denoted by $P_{\text{SLM}}(\cdot; \theta)$. GFlowNets in this setup learn stochastic policies for sampling terminal states by taking actions drawn from the action space. Given an unnormalized reward function $R$ defined over terminal states, the GFlowNets navigates through the state space following the forward policy $P_F(s_{i+1}|s_i; \theta)$. Concretely, for sequence generation, the states $s_i = \text{CONCAT}(c, t_i)$ represent the partial sequence thus far, and action corresponds to a span sampled from the span language model: $a_i \sim P_{\text{SLM}}(a|s_i; \theta)$. Since the action $a_i$ directly determines the next state through span concatenation, we write: $P_F(s_{i+1}|s_i; \theta) = P_{\text{SLM}}(a_i|s_i; \theta)$, where $s_{i+1} = \text{CONCAT}(s_i, a_i)$. This continues until a terminal action $\top$ is sampled, yielding a terminal state $s_n \in \mathcal{X}$ that consists of the prefix, all generated spans, and the termination symbol $\top$.

## 3.2 Learning Objective and Training Policy

We adopt the subtrajectory balance learning objective for training GFlowNets, which, compared to flow matching (Bengio et al., 2021), detailed balance (Bengio et al., 2023) and trajectory balance (Malkin et al., 2022), enables more efficient exploration of DAG-structured state space while providing improved stability, particularly for longer sequence generation (Madan et al., 2023). Given a forward policy $P_F$, a backward policy $P_B$, and a state flow function $F$, the objective over a partial trajectory $\tau = (s_b \to \cdots \to s_e)$ is defined as:

$$\mathcal{L}_{SubTB}(\tau) = \left( \log \frac{F(s_b) \prod_{i=b}^{e-1} P_F(s_{i+1}|s_i)}{F(s_e) \prod_{i=b}^{e-1} P_B(s_i|s_{i+1})} \right)^2. \tag{1}$$

When GFlowNets converge, for a valid state $s_e$ that contains a complete sentence, we define $s_e^\top$ as the terminal state reached after sampling the termination symbol $\top$ from $s_e$. We then have $R(s_e^\top) = F(s_e) P_F(\top \mid s_e)$ for such states (Deleu et al., 2022; Pan et al., 2023; Hu et al., 2024). Using this, we derive the final learning objective by summing over all partial valid trajectories within a complete trajectory $\tau = (s_0 \to s_1 \to \cdots \to s_n)$ with equal weight:

$$\mathcal{L}(\tau) = \sum_{0 \le i < j \le n} \mathbb{I}_{\text{valid}}(s_i, s_j) \left( \log \frac{R(s_i^\top) \prod_{k=i+1}^{j} P_F(s_k|s_{k-1}) P_F(\top|s_j)}{R(s_j^\top) \prod_{k=i+1}^{j} P_B(s_{k-1}|s_k) P_F(\top|s_i)} \right)^2, \qquad (2)$$

where $\mathbb{I}_{\text{valid}}(s_i, s_j)$ is an indicator function that equals 1 if both $s_i$ and $s_j$ are complete sentences, and 0 otherwise. This selective summation focuses the learning signal on transitions between meaningful, complete sentences, leading to more efficient and stable training.

To address the challenge of exploring the combinatorially large sampling space, we propose a hybrid online-offline training approach leveraging readily available training data. While replay buffers enhance GFlowNets training (Jain et al., 2022), these online off-policy methods often suffer from high variance and can become trapped in suboptimal trajectories (Fedus et al., 2020; Vemgal et al.). Drawing inspiration from offline pretraining strategies (Pandey et al., 2025), our approach constructs high-reward trajectories from training data, providing more effective exploration initialization. Specifically, we compose the mini-batch during training using trajectories from three sources: (1) the policy $P_F$, (2) a reward-prioritized replay buffer containing past high-reward trajectories, and (3) trajectories from the training set. This hybrid approach is crucial for FoSS to effectively navigate the combinatorially large DAG-structured state space. Meanwhile, to ensure that the trajectories derived from the training set naturally induce a DAG, we employ a DAG-Inducing span segmentation algorithm that applies a controlled stochastic early-stopping mechanism to standard forward maximum matching. This mechanism results in multiple distinct segmentation trajectories per training set sentence, where different span paths share common sub-sequences, explicitly modeling the DAG structure. For more details, please refer to Appendix E. To further mitigate variance in sparse reward regimes and stabilize training, we follow (Hu et al., 2024) and initially fine-tune the policy network on readily available training set data before standard GFlowNets training, equipping it with the capacity to sample high-reward states.

### 3.3 POLICY NETWORK

We instantiate the forward policy $P_F$ using a span language model that processes the current state $s_i = \text{CONCAT}(c, t_i)$ with a transformer-based prefix encoder, mapping the input to a contextual vector $\mathbf{h}_i \in \mathbb{R}^d$ via causal attention (Vaswani et al., 2017). For the action space, we consider $a \in \mathcal{T} \cup \mathcal{V} \cup \{\top\}$, where $\mathcal{T}$ represents phrases from external corpora and $\mathcal{V}$ is the fixed word-level vocabulary. A span encoder computes context-dependent representations for each potential action $a$, yielding embedding $\mathbf{v}_a \in \mathbb{R}^d$. For phrases from $\mathcal{T}$, we compute vector representations for all candidate phrases from the supporting documents, ensuring the internal state space forms a DAG. We employ a bidirectional Transformer (Devlin et al., 2019) to generate contextualized representations, then transform and concatenate start and end position embeddings through MLPs to form the complete phrase representation. For word-level actions from $\mathcal{V}$ and the terminal action $\top$, we utilize standard token embeddings. Our architecture draws inspiration from recent advances in dynamic vocabulary language models, particularly CoG (Lan et al., 2023), the forward policy distribution over actions is formulated as: $P_{SLM}(a|s_i; \theta) \propto \exp\left(\mathbf{h}_i^\top \mathbf{v}_a\right)$. For the $P_B(s_i|s_{i+1})$, we employed an uniform distribution over all possible suffixes of $s_{i+1}$ in the dynamic vocabulary. Details on dynamic vocabulary construction and policy network are provided in Appendices C and D.

### 3.4 REWARD FUNCTION

Our reward function for span-based text generation balances fluency and alignment with the distribution of human-written text through two components: a language model (LM) and a preference model (PM). The LM component encourages grammatical correctness and contextual relevance (Ide et al., 2024; Fu et al., 2024). However, high-likelihood regions of language model distributions often correspond to repetitive, generic patterns but fail to represent the diversity and quality of human text (Holtzman et al.; Welleck et al., 2020). To mitigate this, our PM component discriminates between human-written text and model-generated outputs, guiding generation toward more human-like expressions. Specifically, the LM component provides a natural measure of textual quality through $p_{\text{LM}}(s_n|c)$, which represents the likelihood of the complete generated sequence given the prefix. For the PM component, our approach employs a Bradley-Terry (Bradley & Terry, 1952) log-likelihood loss with a score-centering regularizer (Eisenstein et al., 2023), which trains the model to assign

Table 1: MAUVE scores on open-ended text generation across in-domain, out-of-domain, and scaling settings. Results are reported for both greedy decoding and nucleus sampling.

| Method | In Domain | | Out of Domain | | Scaling Data Store | |
|---|---|---|---|---|---|---|
| | Greedy | Nucleus | Greedy | Nucleus | Greedy | Nucleus |
| Transformer w/o FT | 18.64 | 22.37 | 20.32 | 25.21 | 19.87 | 23.43 |
| Transformer w/ FT | 19.87 | 23.43 | 23.00 | 26.85 | 20.21 | 21.31 |
| kNN-LM(Khandelwal et al., 2020) | 19.92 | 22.50 | 23.31 | 24.75 | 23.21 | 23.39 |
| RETRO(Borgeaud et al., 2022) | 21.19 | 22.86 | 18.70 | 20.35 | 19.75 | 22.87 |
| CoG(Lan et al., 2023) | 26.01 | 26.14 | 21.31 | 28.14 | 24.68 | 26.97 |
| GFlowNets-FT(Hu et al., 2024) | 26.58 | 29.61 | 26.49 | 28.62 | - | - |
| GDV(Liu et al., 2024) | 25.69 | 24.34 | 26.35 | 24.80 | - | - |
| **FoSS** | **30.78** | **31.65** | **27.84** | **32.17** | **27.88** | **33.79** |

higher scores to human-written references from the training set than to sequences generated by the policy network. The sigmoid-transformed score $p_{\text{PM}}(s_n)$ serves as the learned reward signal.

Combining these complementary objectives, we formulate reward function for a terminal state $s_n$ as:

$$R(s_n) = \exp\left(\alpha \log p_{\text{LM}}(s_n|c) + (1-\alpha) \log p_{\text{PM}}(s_n)\right), \quad \alpha \in (0,1), \tag{3}$$

where $\alpha$ controls the trade-off between language model fluency and preference model alignment. Both models operate at the word-level, requiring tokenization of $s_n = \text{CONCAT}(c, t_n)$ into its constituent tokens $[w_1, w_2, ..., w_N]$, where the first $M$ tokens correspond to the prefix $c$ and the remaining tokens represent the generated text $t_n$ in a total sequence of $N$ tokens. The language model likelihood can then be computed as: $p_{\text{LM}}(t_n|c) = \prod_{i=M+1}^{N} p_{\text{LM}}(w_i|w_1, ..., w_{i-1})$.

For the preference model, we compute $p_{\text{PM}}(s_n) = \sigma(f_{\text{PM}}(w_1, w_2, \ldots, w_N))$, where $f_{\text{PM}}$ represents the discriminator function of the preference model that maps the token sequence to a scalar logit, and $\sigma(\cdot)$ denotes the sigmoid function. A detailed description of the initialization processes for both the LM and PM components is provided in the Appendix C.3.

## 4 EXPERIMENTS

For our experimental evaluation, we compare our approach against base model and state-of-the-art methods: **Transformer** (Vaswani et al., 2017), the de facto standard architecture for neural language modeling; **kNN-LM** (Khandelwal et al., 2020) extends a pre-trained language model by interpolating its predictions with a k-nearest neighbors retrieval, using similar context examples from a datastore to refine next token predictions; **RETRO** (Borgeaud et al., 2022) integrates a frozen BERT (Devlin et al., 2019) retriever with a chunked cross-attention mechanism, allowing the model to condition on retrieved document chunks to predict the next token; **CoG** (Lan et al., 2023) uses a phrase encoder to index the training corpus and a prefix encoder for the current context, retrieving similar phrases based on prefix for sentence continuation, which inspired the architecture of our policy network; **GFlowNets-FT** (Hu et al., 2024) leverages GFlowNets to fine-tune standard Transformers under the token-by-token autoregressive paradigm, resulting in a tree-structured graph $G$; **GDV** (Generation with Dynamic Vocabulary) (Liu et al., 2024) builds upon a similar retrieval framework as CoG, but introduces a novel dynamic vocabulary loss during training to encourage generation beyond the static vocabulary.

To ensure a fair comparison, the prefix encoders in the Transformer, kNN-LM, CoG, GFlowNets-FT, GDV, and FoSS are all initialized from pre-trained GPT-2 (Radford et al., 2019). For the phrase encoder in both CoG and FoSS, we fine-tune the pre-trained BERT-base-cased model (Devlin et al., 2019). In GFlowNets-FT, the reward model employs both the LM and PM components from FoSS. For more implementation details, please refer to the Appendix C. We report baseline results from their original publications where possible.

Table 2: Pairwise comparison of FoSS against baselines using GPT-4 as evaluator for text quality assessment.

| In Domain | | | | Out of Domain | | | |
|---|---|---|---|---|---|---|---|
| **FoSS vs.** | **GPT4 Preference** | | | **FoSS vs.** | **GPT4 Preference** | | |
| | **Better** | **Neutral** | **Worse** | | **Better** | **Neutral** | **Worse** |
| **Transformer** | **53%** | 19% | 28% | **Transformer** | **56%** | 23% | 21% |
| **kNN-LM** | **67%** | 15% | 18% | **kNN-LM** | **75%** | 13% | 12% |
| **CoG** | **42%** | 31% | 27% | **CoG** | **74%** | 6% | 20% |
| **GFlowNets-FT** | **55%** | 29% | 16% | **GFlowNets-FT** | **49%** | 24% | 27% |

Following previous work (Su et al., 2022; Liu et al., 2024; Li et al., 2024), we evaluate models using **MAUVE** (Pillutla et al., 2021) and **Diversity** (Welleck et al., 2020) metrics. MAUVE measures the divergence between model-generated and human-written text distributions using a quantized embedding space and Diversity quantifies the non-repetitive nature of generated content through n-gram statistics. Further details on the evaluation metrics are provided in Appendix B. For decoding, we use greedy search, which selects the highest-probability span at each step, and nucleus sampling, which samples the next span from the normalized probability distribution with top_p set to 0.95 (Holtzman et al.).

### 4.1 IN DOMAIN EVALUATION

For in-domain evaluation, models are trained on the training set of the WikiText-103 (Merity et al., 2016) dataset and evaluated on its test set. The WikiText-103 dataset is a large-scale corpus comprising 1,801,350 training samples, 3,760 development samples, and 4,358 test samples derived from Wikipedia articles, which serves as a standard benchmark for assessing language modeling performance (Su et al., 2022; Li et al., 2024).

Table 1 presents the performance comparison between the baselines and our proposed FoSS on the WikiText-103 test set. Our FoSS achieves MAUVE score absolute improvements of 5.51 and 8.22 over CoG and fine-tuned Transformer with nucleus sampling, respectively. This improvement can be attributed to GFlowNet's ability to model DAG-structured internal state space and generalize better. Meanwhile, FoSS outperforms GFlowNets-FT by 2.04, highlighting the advantage of our DAG-structured state space over the tree-structured approach used in GFlowNets-FT, which enables more comprehensive exploration of state transitions. Notably, despite the established tendency of greedy search to produce degeneration problems (Welleck et al., 2020), FoSS with greedy search outperforms both the fine-tuned Transformer and the CoG with nucleus sampling by 7.35 and 4.64 in MAUVE, respectively.

To further evaluate the quality of generated text, we employ GPT-4 to assess the continuations generated by the models for each test prefix. Prior work has shown that powerful LLMs, such as GPT-4, are capable of matching human preferences well, achieving over 80% agreement, which is comparable to the agreement between humans (Zheng et al., 2023). Our evaluation focuses on key aspects of text quality: fluency, coherence, informativeness, and grammatical correctness. Further details can be found in the Appendix G. As shown in Table 2, the GPT-4 evaluation results reveal that our FoSS model consistently outperforms all baselines across all quality dimensions.

### 4.2 OUT OF DOMAIN EVALUATION

In the domain adaptation setting, the models trained on the WikiText-103 dataset are tested on a different domain. Following previous work(Liu et al., 2024; Cao et al., 2024), we use the English portion of Law-MT (Koehn & Knowles, 2017), which is an English-German translation dataset in the legal domain containing 389,292 training samples with 2,000 samples each for development and testing purposes. The memory of kNN-LM, RETRO, CoG, GDV and FoSS are constructed from the training set of Law-MT. Transformer w/o ft is fine-tuned on the WikiText103 training set, and Transformer w/ ft is additionally trained on the Law-MT training set.

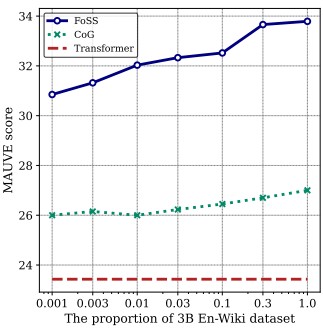 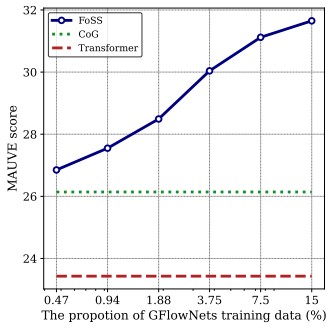 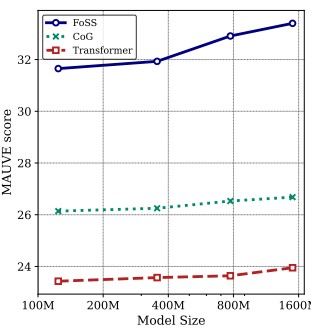

Figure 2: Generation quality of FoSS with different sizes of the span index.

Figure 3: Generation quality of FoSS with different sizes of offline data trained.

Figure 4: Generation quality of FoSS with different sizes of models.

As shown in Table 1, FoSS consistently outperforms all baseline methods, including the Law-MT fine-tuned Transformer. Specifically, FoSS achieves MAUVE score improvements of 6.96 and 4.03 over the fine-tuned Transformer and CoG, respectively. These results demonstrate that FoSS effectively adapts to new domains by simply updating the source text collection without requiring domain-specific training. The superior performance can be attributed to the generalization capabilities of GFlowNets, which enable more effective knowledge transfer across domains.

Additionally, we conducted a GPT-4 evaluation under this setting, which follows a similar procedure as outlined in Section 4.1. As presented in Table 2, FoSS-generated continuations were preferred in 56% and 74% of cases when compared to the fine-tuned Transformer and CoG, respectively. These human-aligned evaluation results demonstrate the effectiveness of FoSS in cross-domain generation scenarios without requiring domain-specific parameter updates.

## 4.3 SCALING EVALUATION

In this section, we evaluate the scaling properties of FoSS across three dimensions: memory size, training data volume, and model scale. For memory scaling, we constructed retrieval datastores for kNN-LM, RETRO, CoG, and FoSS from the En-Wiki corpus and evaluated on the WikiText-103 test set. The Transformer w/o FT was trained solely on WikiText-103, while Transformer w/ FT was additionally trained on the En-Wiki.

As shown in Table 1, FoSS with the En-Wiki memory consistently outperforms other strong baselines, including FoSS with the smaller WikiText-103 memory. This demonstrates the plug-and-play capability of FoSS, which can leverage larger memory resources for performance gains without requiring additional training. To further investigate the impact of memory size on generation quality, we created subsets of En-Wiki at varying scales to serve as the retrieval datastore. Figure 2 illustrates that as we increase the FoSS memory size, text generation quality consistently improves. For comparative purposes, we use Transformer w/o FT as the baseline, since the fine-tuned Transformer exhibited performance degradation when trained on the additional En-Wiki data.

To assess the impact of training data volume on FoSS performance, we sampled subsets of the WikiText-103 training set at different scales and trained both models accordingly, evaluating them on the test set. As illustrated in Figure 3, FoSS outperforms fully-trained CoG and Transformer models even when trained on an extremely limited data fraction (0.47%). Furthermore, FoSS demonstrates robust scaling behavior, with performance consistently improving as training data volume increases. This finding highlights the sample efficiency of our approach and suggests that FoSS effectively leverages the compositional nature of the DAG-structured state space to generalize from limited examples, while maintaining strong scaling properties with additional training data.

Finally, to evaluate model scaling effects, we initialized the prefix encoder with progressively larger versions of GPT-2: GPT-2 (base), GPT-2-Medium, GPT-2-Large, and GPT-2-XL. All models were trained on WikiText-103 and evaluated on its test set. Figure 4 shows that text generation quality consistently improves as we scale up the model size, confirming that FoSS effectively leverages increased model capacity.

Table 3: Accuracy of Different Methods on Knowledge-Intensive Tasks

| Method | TruthfulQA | OpenbookQA | ARC-Challenge | MedMCQA | Med-USMLE |
|---|---|---|---|---|---|
| Transformer (w/o FT) | 28.02 | 22.67 | 23.82 | 26.70 | 24.89 |
| Transformer (w FT) | 28.76 | 22.71 | 24.00 | 26.95 | 24.15 |
| CoG | 29.38 | 24.29 | 24.34 | **27.55** | 25.06 |
| **FoSS** | **30.45** | **26.20** | **24.63** | 27.44 | **25.27** |

## 4.4 DOWNSTREAM EVALUATION

Consistent with prior research (Sanh et al., 2021; Brown et al., 2020), we adopt a classification-with-options approach to evaluate model performance. For assessment, we employ five diverse knowledge-intensive datasets: TruthfulQA (Lin et al., 2022) for evaluating factual accuracy, Open-BookQA (Mihaylov et al., 2018) and ARC-Challenge (Clark et al., 2018) for testing scientific reasoning at different complexity levels, and MedMCQA (Pal et al., 2022) and Med-USMLE (Jin et al., 2021) for assessing specialized medical knowledge.

In this framework, the model is presented with a set of candidate options, and the likelihood of each option being the correct answer is computed. The option with the highest probability is selected as the model's prediction. We then report the accuracy of the model's predictions. To calculate the likelihood of a given text, we approximate the likelihood by summing over all possible generation paths using dynamic programming (Cao et al., 2024).

Table 3 presents the accuracy comparisons between the baselines and our proposed FoSS on five question-answering datasets. The Transformer w/ FT baseline was fine-tuned on the WikiText-103 dataset. Our results show that the proposed FoSS consistently outperforms the baselines across most datasets. Specifically, compared to the standard Transformer w/ FT, our model improves accuracy on the TruthfulQA and OpenBookQA datasets, increasing from 28.76% to 30.45% and from 22.71% to 26.20%, respectively. While CoG shows competitive performance on MedMCQA, our method still achieves the best overall performance across the majority of the benchmarks.

## 4.5 ABLATION STUDY

Table 4: Ablation study on key components of FoSS: DAG structure, preference model (PM), and language model (LM) rewards.

| Method | In Domain | | Out of Domain | | Scaling Data Store | |
|---|---|---|---|---|---|---|
| | Mauve | Diversity | Mauve | Diversity | Mauve | Diversity |
| **FoSS** w/o DAG | 29.61 | 65.72 | 28.62 | 81.44 | - | - |
| **FoSS** w/o PM | 28.25 | 89.91 | 30.49 | 92.53 | 29.81 | 91.42 |
| **FoSS** w/o LM | 29.09 | **92.77** | 29.79 | **94.72** | 29.53 | 92.46 |
| **FoSS** | **31.65** | 92.48 | **32.17** | 93.00 | **33.79** | **92.51** |

To validate the effectiveness of modeling $G$ as a general DAG rather than a tree, we conducted an experiment where we removed all phrases from the vocabulary during training, ensuring that $G$ forms a tree structure. As shown in Table 4, the results demonstrate that FoSS significantly outperforms the tree-structured variant across all evaluation settings. Specifically, the DAG-based model achieves a 9.65% relative improvement in MAUVE score and a 27.46% increase in Diversity score across different settings. This indicates that the DAG structure allows for more diverse and flexible generation paths, enabling better exploration of the generation space and mitigating the limitations posed by tree-structured generation, which restricts the ability to explore diverse phrase combinations.

To assess the impact of different reward components, we conducted ablation experiments by separately removing the LM and PM from Equation 3. As demonstrated in Table 4, removing either component

leads to decreased MAUVE scores across all settings, indicating that both components are essential and complementary. It is worth noting that the PM-only variant (FoSS w/o LM) achieves the highest Diversity scores in both in-domain and out-of-domain settings, suggesting that PM inherently favors text with greater diversity. This aligns with PM's objective to differentiate human-written text, which typically exhibits richer diversity from model-generated texts (Zhang et al., 2025).

## 5 CONCLUSION

In this work, we construct a dynamic span-vocabulary for text generation, framing the state space as a DAG. We then model the span generation within the GFlowNets framework, unleashing the potential of GFlowNets in language models within the DAG-structured space. Our proposed FoSS excels in text generation, domain adaptation, and knowledge-intensive tasks, consistently outperforming strong baseline models. The scaling experiment further demonstrated that FoSS consistently improves in performance with increases in model parameters, training data, and retrieval data, validating its scalability and potential with greater computational budgets.

## ACKNOWLEDGMENTS

This work is sponsored by the National Natural Science Foundation of China (NSFC) grant (No.62525209, 62576211).

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

# A    RELATED WORK

## A.1    MODELS BEYOND FIXED VOCABULARY

Recent advancements in language modeling have shifted towards dynamic vocabularies and flexible text spans to address the limitations of traditional token-by-token autoregressive generation. Retrieval-based methods have gained significant attention, with approaches like CoG-2 (Cao et al., 2024) and CoG (Lan et al., 2023) leveraging external datastores to dynamically retrieve and assemble contextually relevant phrases. This allows for a flexible vocabulary that enhances both semantic richness and fluency in generated text. Similarly, chunk-based methods, such as the kNN machine translation model by (Martins et al., 2022), retrieve variable-length token sequences, improving translation coherence and fluency. Speculative decoding techniques, like CopySpec (Dumitru et al., 2025), further optimize this idea by reusing repeated token sequences, which accelerates inference and reduces redundancy. Domain-specific enhancements, such as those proposed by (Giannone et al.), dynamically inject specialist tokens into the model's vocabulary, enabling more effective handling of domain-specific knowledge without the need for extensive fine-tuning. In the realm of semi-parametric methods, NEST (Li et al., 2024) introduces text span retrieval from a corpus, allowing for more diverse and semantically coherent text generation. Additionally, (Liu et al., 2024) propose a dynamic vocabulary approach that incorporates arbitrary text spans as fundamental generation units, demonstrating improvements in both generation quality and efficiency across various tasks. Collectively, these approaches represent a paradigm shift from static token generation to more flexible, context-aware methods, enabling language models to produce semantically richer and more diverse content.

However, while methods such as (Lan et al., 2023), (Cao et al., 2024), and (Liu et al., 2024) implicitly induce a DAG-structured internal state space (each state can be reached via two segmentation paths), this structural property remains unrecognized and unexploited. Consequently, existing approaches exhibit biases towards specific segmentation patterns. Moreover, thoroughly exploring and generalizing over the DAG-structured state space is inherently nontrivial, as it requires exponentially growing data to sufficiently cover the combinatorial explosion of compositional paths. To address this challenge, we identify GFlowNets as a principled solution, enabling effective generalization and exploration in DAG-structured spaces (Shen et al., 2023; Krichel et al., 2024; Atanackovic & Bengio, 2024).

## A.2    GENERATIVE FLOW NETWORKS

Generative Flow Networks (GFlowNets) offer a framework for training stochastic policies to sample compositional objects with probabilities proportional to a reward function (Bengio et al., 2023). This unique property makes them particularly effective in applications that require generating diverse yet high-reward samples, such as biological sequence design (Jain et al., 2022; M Ghari et al., 2024; Kim et al.) and molecule generation (Koziarski et al., 2024; Lu et al., 2024). Recently, GFlowNets have also been successfully extended to various inference tasks, including Bayesian structure learning (Shen et al.) and variational inference with discrete latent variables (Hu et al., 2023). In the context of language modeling, (Hu et al., 2024) introduced a method to amortize intractable inference in large language models using GFlowNets, demonstrating its potential in guiding language model behavior. Subsequently, (Yu et al., 2025) utilized GFlowNets to generate diverse reasoning paths, enhancing the exploration of solutions in complex problem-solving scenarios. Additionally, recent studies have explored the use of GFlowNets for automated red-teaming of large language models. For instance, (Lee et al., 2025) proposed a two-stage approach combining GFlowNets fine-tuning and MLE smoothing to generate diverse adversarial prompts, mitigating mode collapse issues typical in reinforcement learning-based methods.

The original motivation for the GFlowNets machinery, and its primary novelty, are in MDPs with many trajectories per state (Shen et al., 2023; Bengio et al., 2021). However, existing GFlowNets approaches integrated with LLMs remain constrained by token-level generation. Consequently, they inherently form tree-structured state spaces, which reduces the GFlowNets framework to a degraded learning paradigm that limits the exploration of state transition. In contrast, our method constructs a general DAG by employing text spans as actions, allowing for multiple generation paths to the same state, thus fully leveraging the exploration and generalization potential of GFlowNets.

# B DATASETS AND METRICS DETAILS

## B.1 DATASETS DETAILS

The open-ended text generation experiments in this paper include three benchmarks: (1) WikiText-103, a large-scale corpus comprising 1,801,350 training samples, 3,760 development samples, and 4,358 test samples derived from Wikipedia articles, containing over 100 million words and widely used for evaluating universal language modeling performance; (2) the English portion of Law-MT, an English-German translation dataset in the legal domain containing 389,292 training samples with 2,000 samples each for development and testing purposes; and (3) En-Wiki, an extensive corpus utilized for enlarged phrase indexing experiments, containing over 4,848,348 long English Wikipedia documents with more than 3 billion words, substantially larger than WikiText-103.

Given that En-Wiki serves as the retrieval datastore in the scaling experiments, we examined the overlap between En-Wiki and the WikiText-103 test set. Empirically, only $8.5 \times 10^{-6}$ of En-Wiki documents appear in the WikiText-103 test set. When the retriever queries En-Wiki using prefixes from the WikiText-103 test samples and retrieves 1,024 documents per prefix, only 8 test samples have any retrieved document containing text identical to the corresponding test sample.

For knowledge-intensive evaluation tasks, we utilize several established benchmarks: (1) Open-BookQA (Mihaylov et al., 2018), which evaluates understanding of scientific concepts through 5,957 multiple-choice elementary science questions (4,957 train, 500 development, 500 test) and probes 1,326 core scientific facts requiring broad commonsense knowledge—we utilize its test split comprising 500 questions; (2) ARC-Challenge (Clark et al., 2018), containing 7,787 grade-school level, multiple-choice science questions across diverse topics, with its Challenge Set specifically including questions answered incorrectly by retrieval-based and word co-occurrence algorithms—we use the test split of this Challenge Set, consisting of 1,172 questions; (3) TruthfulQA (Lin et al., 2022), a benchmark designed to evaluate the truthfulness of language models, consisting of 817 multiple-choice questions across 38 categories including health, law, finance, and politics; (4) MedMCQA (Pal et al., 2022), comprising 194,000 multiple-choice questions covering 21 medical subjects and 2,400 healthcare topics—we use its validation split consisting of 4,183 questions; and (5) Med-USMLE (Jin et al., 2021), which comprises 12,723 multiple-choice questions with four options each, sourced from the United States Medical Licensing Examination—we utilize its test split containing 1,273 questions.

## B.2 METRICS DETAILS

**MAUVE** (Pillutla et al., 2021) measures the divergence between model-generated and human-written text distributions using a quantized embedding space. By computing the divergence between discrete distributions derived from neural embeddings, MAUVE effectively captures both quality and diversity aspects of generated text, correlating strongly with human judgments while remaining computationally efficient.

**Diversity** quantifies the non-repetitive nature of generated content through n-gram statistics. Formally defined as $\Pi_{n=2}^{4}(1 - \frac{\text{Rep}-n}{100})$, where $Rep-n$ represents the percentage of duplicate n-grams, this metric penalizes redundant patterns while rewarding informative and varied text. Higher Diversity scores indicate more information-rich generations with fewer repetitive structures.

# C IMPLEMENTATION DETAILS

## C.1 TRAINING SETUP AND HYPERPARAMETERS

The training of our proposed model was conducted on eight NVIDIA A800 GPUs, each equipped with 80GB of memory. We employed the AdamW optimizer with a learning rate of 1e-7 and a linear learning rate schedule. To manage memory constraints, we implemented gradient accumulation with a step size of 32. For the first epoch, training relied exclusively on offline training set data. Starting from the second epoch, at each step, we sampled trajectories with a probability of 0.2 from the replay buffer (offline) and with a probability of 0.8 from online-generated trajectories. For the language model, we followed the configuration in (Hu et al., 2024), applying LoRA (Hu et al., 2022)

for fine-tuning GPT-2 and BERT with the following hyperparameters: rank $r = 64$, scaling factor $\alpha = 16$, and LoRA dropout rate of 0.1.

## C.2 DYNAMIC VOCABULARY CONSTRUCTION

To enhance the efficiency of FoSS, we pre-encoded all documents in the source text collection offline. However, retrieving from such an extensive phrase collection poses significant engineering challenges. To address this, we adopted a coarse-to-fine pipeline, consistent with the approach in CoG. Specifically, during training, the dynamic vocabulary of FoSS comprises two components: (1) a token-level vocabulary and (2) phrases derived from the batch of training documents, augmented by the top-$k$ documents (with $k = 128$) that share similar topics with the prefix of the training documents. For the second component, we selected all substrings of length 2-8 tokens from the documents, ensuring that the policy network can sample any sequence in an exponentially scalable manner. Notably, experiments revealed that relying solely on the batch of training documents, which were obtained through the span segmentation algorithm, resulted in poor-quality online-sampled sentences that adversely affected training stability. Additionally, during the FoSS training, we needed to segment prefixes from training samples before trajectory sampling. In our experiments, we first used NLTK's sentence tokenizer to split training samples into sentences, then progressively concatenated these sentences to the prefix until its length approached 32 tokens, which matches the prefix length used during testing. Due to computational resource constraints, we sampled 15% of the data for GFlowNets training. During inference, we first employed a document retriever to identify the top-$k$ related documents for a given prefix, with $k = 1024$. Subsequently, the corresponding phrase representations were collected for generation. In this work, we utilized a widely adopted semantic matching model, DPR (Karpukhin et al., 2020), combined with the FAISS (Johnson et al., 2019) vector search toolkit, as the document retriever to recall documents with topics similar to the prefix.

## C.3 REWARD MODEL TRAINING

For both the LM and PM components of the reward function, we utilized GPT-2 for full parameter fine-tuning. In the case of LM fine-tuning, we employed causal language modeling loss on the training dataset. For the PM fine-tuning, we employ Bradley-Terry objective that encourages the model to assign a higher score to the preferred continuation than to its non-preferred counterpart. We train on a dataset $\mathcal{D}_{\text{PM}} = \{(x^+, x^-)\}$, where $x^+$ is the human-written references from the training set and $x^-$ the continuations produced by the initial policy. Let $f_{\text{PM}}(x)$ denote the scalar-valued score predicted by the PM. Our training objective is:

$$\mathcal{L}_{\text{PM}} = -\mathbb{E}_{(x^+,x^-)\sim\mathcal{D}_{\text{PM}}}\left[\log \sigma(f_{\text{PM}}(x^+) - f_{\text{PM}}(x^-) - m)\right] + \lambda \, \mathbb{E}_{(x^+,x^-)\sim\mathcal{D}_{\text{PM}}}\left[(f_{\text{PM}}(x^+) + f_{\text{PM}}(x^-))^2\right],$$

where $\sigma(\cdot)$ is the sigmoid function, $m$ represents an optional margin hyperparameter and $\lambda$ controls the strength of the centering regularization. Here, we employ the score-centering regularizer (Eisenstein et al., 2023), adding an auxiliary loss that minimizes the squared sum of the scores. This encourages the model to produce mean-zero outputs.

## D POLICY NETWORK DETAILS

Our policy network architecture comprises two primary components: a prefix encoder and a span encoder. The prefix encoder processes the current state as a sequence of tokens, where previously sampled spans are tokenized. This sequence is then encoded using a standard Transformer architecture with causal attention mechanisms. The representation of the prefix is derived from the final layer's hidden state corresponding to the last token in the sequence.

For the span encoder, we compute vector representations for all candidate phrases from the supporting documents (constituting $\mathcal{T}$). Specifically, we extract all possible substrings of length 2-8 tokens from the supporting documents to form $\mathcal{T}$, ensuring that the internal state space forms a DAG. To encode these phrases, we employ a deep bidirectional Transformer (Devlin et al., 2019) to generate contextualized token representations for each supporting document. For a phrase spanning from position $s$ to $e$, we apply separate MLPs to transform token representations into start and end

embeddings of dimension $\frac{d}{2}$ each, then concatenate the start embedding at position $s$ with the end embedding at position $e$ to form the complete phrase representation. For single words in $\mathcal{V}$ and the terminal action $\top$, we utilize standard token embeddings.

## E    SPAN SEGMENTATION ALGORITHM

In this work, we introduce a DAG-Inducing Probabilistic Span Segmentation algorithm that generates stochastically varied segmentation trajectories for each training document, specifically designed to facilitate the offline construction of DAGs over sequences. Our span segmentation algorithm draws inspiration from CoG. Building upon the standard forward maximum matching strategy, our method integrates a probabilistic early-stopping mechanism that allows phrase extraction to terminate at controlled random points, governed by a set of thresholds $\mathcal{P} = \{p_0 = 0, p_1, \ldots, p_n \mid 0 \leq p_i < 1, \forall i = 0, \ldots, n\}$.

Concretely, given a document, we tokenize it into a sequence of tokens and iteratively scan from left to right. At each step, we attempt to find the longest token span that appears in either the document itself or its top-k retrieved nearest neighbors. If a candidate span is found and its length falls within pre-specified bounds, it is selected as a phrase with probability $p_r$, where $p_r \in \mathcal{P}$. If selected, we commit this candidate span as a phrase in the segmentation and reset our span search to begin at the next token. If not selected, we continue the segmentation process by extending the current span with one additional token, repeating this procedure until either we find a selectable span, reach the maximum span length limit, or the extended span no longer appears in the retrieved documents. The details of our proposed DAG-Inducing Probabilistic Span Segmentation algorithm can be found in Algorithm 2. This probabilistic termination mechanism results in multiple distinct segmentation trajectories per document, where different span paths share common sub-sequences. Crucially, by generating overlapping and diverging segmentations offline, we ensure that the resulting training trajectories data naturally induce a DAG structure.

## F    EFFICIENCY AND DIVERSITY ANALYSIS

We compare the average time cost of different methods for completing the generation on the test set following the speed testing setup of Lan et al. (2023). The results are reported in Table 5. As can be seen, FoSS achieves comparable inference efficiency with the standard Transformer baseline. This is because the copied phrases typically consist of multiple tokens. As a result, FoSS requires fewer decoding steps to generate text of the same length. Unlike FoSS, which uses a coarse-to-fine search pipeline, kNN-LM performs large-scale vector search at every decoding step. This leads to significantly higher inference latency compared to other methods.

Regarding training efficiency, FoSS incurs additional computational costs compared to offline training methods due to its online sampling mechanism, which is inherent to RL-based approaches. The most comparable baseline in terms of training paradigm is GFlowNets-FT (Hu et al., 2024). On 8 GPUs, FoSS requires 10.5 hours per epoch, compared to 9.25 hours for GFlowNets-FT. This modest overhead stems from the expanded vocabulary needed to explore the DAG-structured state space, which enables more comprehensive exploration of compositional paths.

Meanwhile, we also compare the diversity of the methods. It can be observed that FoSS shows improvements in diversity compared to CoG under both generation strategies, with the diversity score increasing from 43.03 to 43.50, and from 89.07 to 92.48, respectively. This enhancement can be attributed to the fundamental training objective of GFlowNets, which learns a policy that samples from a distribution proportional to the reward function ($P_F^\top(x) \propto R(x)$) rather than simply maximizing the reward, which encourages exploration of diverse generation trajectories and enhances the diversity of the generated text. While the diversity of our method under nucleus sampling is slightly lower than kNN-LM, this can be attributed to differences in the inference strategy. kNN-LM conducts large-scale vector search at every decoding step, whereas FoSS performs a single retrieval step during the generation process, which affects the richness of the retrieved content and subsequently impacts the overall diversity of the generated text.

**Algorithm 2:** DAG-Inducing Probabilistic Phrase Segmentation

---

**Data:** Document set: $\mathcal{D} = \{d_i, \{d_j\}_{j=1}^{K}\}_{i=1}^{N}$, where $d_i$ denotes the $i$-th document. $K$ denotes the number of retrieved documents. $N$ denotes the number of documents in the training set. Pre-defined maximum and minimum phrase lengths: $L_{max}$ and $L_{min}$. Phrase segmentation thresholds: $\mathcal{P} = \{p_0 = 0, p_1, \ldots, p_n \mid 0 \leq p_i < 1, \forall i = 0, \ldots, n\}$.

**Result:** DAG-inducing segmented document set:

$\mathcal{D}' = \{\{\mathcal{V}_r^{(i)} = \{(p_{i,x}^{(r)}, (d_j, \text{pos}_j))\}_{x=1}^{||d_i||_p^{(r)}}\}_{r=0}^{n}\}_{i=1}^{N}$,

where $p_{i,x}^{(r)}$ denotes the $x$-th phrase in the $r$-th segmentation of $d_i$, appearing in document $d_j$.

**Preprocess**: Split each document into token-level sequences using an off-the-shelf tokenizer. Define empty result set $\mathcal{D}' = \{\}$.

---

**for** $i \leftarrow 1$ **to** $N$ **do**
    **for** $p_r \in \mathcal{P}$ **do**
        cursor $\leftarrow 0$
        PhraseCollection $\leftarrow \{\}$
        cache$_p \leftarrow \{\}$
        label$_{last} \leftarrow$ False
        **while** *cursor* $\leq ||d_i||_t$ **do**
            **if** $L_{min} \leq len(cache_p) \leq L_{max}$ **then**
                **if** $random()$ $\geq p_r$ **then**
                    label$_{now}$, rest $\leftarrow$ SearchPhrase (cache$_p$)
                **else**
                    label$_{now}$, rest $\leftarrow$ False, $\{\}$
            **else**
                **if** $len(cache_p) > L_{max}$ **then**
                    cache$_p \leftarrow \{\}$
            **if** *label$_{last}$ = True **and** label$_{now}$ = False* **then**
                cursor $\leftarrow$ cursor $-1$
                PhraseCollection.append(cache$_p$, rest)
                cache$_p \leftarrow \{\}$
            **else**
                **if** *label$_{last}$ = False **and** label$_{now}$ = False* **then**
                    PhraseCollection.append(cache$_p$, None)
                    cache$_p \leftarrow \{\}$
            cursor $\leftarrow$ cursor $+1$
            label$_{last} \leftarrow$ label$_{now}$
        $\mathcal{D}'[i].append$(PhraseCollection)

Table 5: Comparison of Diversity and Latency Among Methods for In Domain Generation

| Method | Greedy | | Nucleus | |
|---|---|---|---|---|
| | Diversity↑ | Latency(s)↓ | Diversity↑ | Latency(s)↓ |
| Transformer | 22.37 | 1.32 | 93.22 | **1.48** |
| kNN-LM | 22.13 | 10.36 | **95.80** | 10.42 |
| RETRO | 21.19 | 4.39 | 91.19 | 4.51 |
| CoG | 43.03 | 1.29 | 89.07 | 1.54 |
| **FoSS** | **43.50** | **1.29** | 92.48 | 1.51 |

---

**Case Prefix**

The production selected two adjacent properties on the Warner backlot's 'Blondie Street' for the Burnham and Fitts' homes. The crew

**Generation of FoSS**

continued filming at the same location over several years, aiming to maintain continuity for scenes set at the school. The producers noted that filming would need several years to complete and that it was easier to edit the film and prepare it for the same location used for other scenes in the film.

**Generation of Transformer w/ FT**

moved there in 1950, using three yards of undeveloped land : Hawthorne Street between 9 and 10 Westheimer Street ( the site of the National Training Center at that time ), a district section of South Main Street between 14th and 15th Streets, and the neighborhood on North College Street opposite Latimer streets.

---

Figure 5: Case Study. The blue part represents directly sampling a phrase.

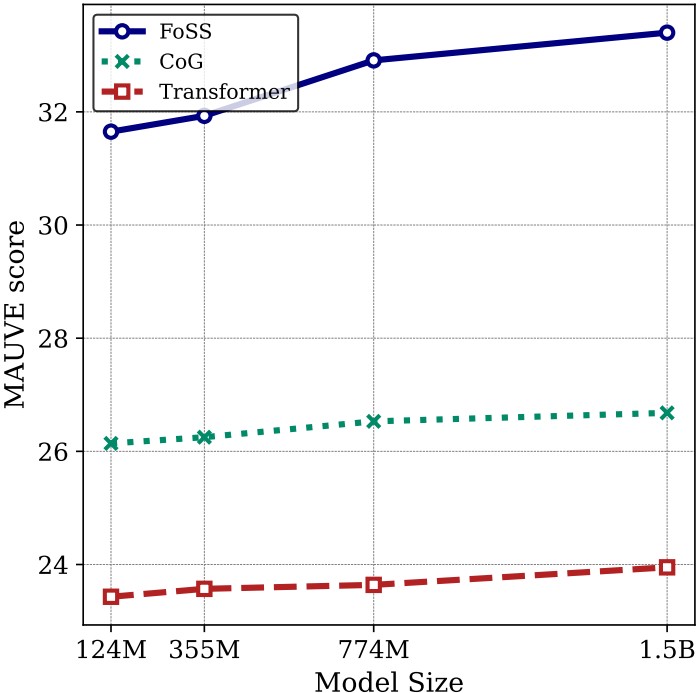

Figure 6: Generation quality of FoSS with different sizes of models.

## G  GPT4 EVALUATION

Due to GPT-4's sensitivity to the order of candidate sentences (Wang et al., 2024), we adopt the method described in (Wang et al., 2024; Liu et al., 2024), computing the final result by averaging outcomes from pairs of evaluations with reversed presentation order. Figure 7 shows the detailed prompt used for GPT-4, which we adopt from (Liu et al., 2024). Specifically, for each triplet (prefix, generation_1, generation_2), we include another corresponding triplet (prefix, generation_2, generation_1), in order to mitigate any impact that the order of the two generations might have on GPT-4's evaluation.

**System:** You are a helpful and precise assistant for checking the quality of the text.

**User:** [Prefix]
{Prefix of the Test Sample}
[The Start of Assistant 1's Generation]
{Generation_1}
[The End of Assistant 1's Generation]
[The Start of Assistant 2's Generation]
{Generation_2}
[The End of Assistant 2's Generation]
[System]
We would like to request your feedback on the performance of two AI assistants in response to the user prefix displayed above. Please rate the fluency, coherence, informativeness, and grammar. Each assistant receives an overall score on a scale of 1 to 10, where a higher score indicates better overall performance. Please first provide a comprehensive explanation of your evaluation, avoiding any potential bias and ensuring that the order in which the responses were presented does not affect your judgment. Then, output two lines indicating the scores for Assistant 1 and 2, respectively.
Your response should be in the following \*\*json\*\* format:
```json
{{
    "Evaluation evidence": "your evaluation explanation here",
    "Score of the Assistant 1": "score",
    "Score of the Assistant 2": "score"
}}```

Figure 7: The GPT-4 evaluation template with three slot {Prefix of the Test Sample}, {Generation_1}, {Generation_2}

## H CASE STUDY

Figure 5 presents a comparative example between FoSS and a fine-tuned GPT2 model. Given an identical film production prefix, FoSS generates a continuation that maintains thematic coherence, discussing filming logistics, continuity considerations, and production planning. The text flows naturally from the established context about Warner's backlot filming. In contrast, the GPT2 model produces text that, while grammatically correct, diverges semantically with abrupt shifts to specific street locations and unrelated elements like "National Training Center" that lack connection to the film production theme.

## I LIMITATIONS

One limitation of FoSS is its design to perform a single retrieval operation per prefix, in contrast to token-level retrieval strategies (e.g., kNN-LM). While this approach yields substantial gains in inference latency (as detailed in Table 5), the comparatively lower generation diversity observed against such baselines is an anticipated trade-off. Future work could explore augmenting retrieval frequency to potentially enhance diversity and overall performance; however, this direction is orthogonal to the primary methodological contributions of this study and is thus reserved for subsequent investigation.

## J THE USE OF LLMS

We used LLMs for minor language polishing to improve clarity and readability. Additionally, we used GPT-4 to evaluate the quality of model-generated texts, as described in the Experiments section.

