# OpenReview forum: "Flow of Spans: Generalizing Language Models to Dynamic Span-Vocabulary via GFlowNets"
_ICLR.cc/2026/Conference — ICLR 2026 Poster_

### Official Review · Reviewer_z7qh · 2025-10-28

**Soundness:** 2
**Presentation:** 3
**Contribution:** 3
**Rating:** 6
**Confidence:** 3

**Summary:**

This paper proposes FoSS (Flow of SpanS), a span-based language generation framework that models generation as trajectories on a DAG (instead of the usual token-level tree) and trains a span-generating policy with Generative Flow Networks (GFlowNets). The authors introduce a DAG-inducing probabilistic span segmentation to produce multiple segmentation trajectories per sentence, construct a dynamic span vocabulary from retrieved documents, and instantiate a span policy (prefix encoder + span encoder) trained with a subtrajectory-balance GFlowNet objective. A combined reward (fine-tuned LM likelihood + a preference model) steers generation toward fluent, human-like outputs. Empirically, FoSS reports substantial MAUVE gains over strong baselines (CoG, kNN-LM, RETRO, Transformer variants) and wins in GPT-4 preference comparisons; ablations show the DAG design and reward components are important.

**Strengths:**

1. Clear methodological novelty — DAG + GFlowNets for spans.
Modeling span selection as actions yields a DAG where the same final string can be composed via multiple span paths; using GFlowNets to learn a forward policy which samples terminal sequences with probability proportional to reward is a principled fit for such DAG-structured compositional spaces. This is a clean conceptual advance over token-level GFlowNet applications.



2. Practical span segmentation and dynamic vocabulary.
The DAG-inducing probabilistic span segmentation (probabilistic early stopping on forward maximum matching) creates multiple offline trajectories per document and makes the DAG explicit for training. The dynamic vocabulary construction (token vocab + substrings from retrieved documents, 2–8 tokens) is a sensible engineering choice to make the action space manageable while expressive.

3. Thoughtful hybrid training recipe.
The hybrid online/offline batching (policy samples + reward-prioritized replay + training-derived trajectories) and pre-fine-tuning of policy components to stabilize sparse-reward GFlowNet training are appropriate design choices for the large combinatorial space. Algorithm 1 clearly describes the training loop.


4. Extensive empirical evaluation.
The paper reports (i) MAUVE and Diversity on WikiText-103 and Law-MT, (ii) GPT-4 preference evaluations, (iii) downstream knowledge tasks (TruthfulQA, OpenBookQA, ARC, MedMCQA, Med-USMLE), (iv) scaling experiments (memory, training data fraction, model size), and (v) ablations showing the DAG and reward components matter. The breadth of evaluation supports the central claims.

**Weaknesses:**

1. Reward design — potential for reward hacking / calibration issues.
The reward R = exp(α log p_LM + (1−α) log p_PM) combines LM likelihood and a learned preference model. Preference models can be brittle and susceptible to distributional shift / reward hacking; the paper briefly mentions score-centering regularizer but lacks a systematic analysis of sensitivity to α, PM training data composition, or adversarial failures. More diagnostics would strengthen confidence.

2. Computational cost / scalability concerns not fully quantified.
Training uses eight A800 GPUs with gradient accumulation and LoRA; the span index and retrieval (up to k=1024 at inference) plus GFlowNet trajectory sampling could be expensive. The latency table (Table 5) is referenced, but details are not shown in the snippet; a clearer cost vs. baseline comparison (GPU hours, wall-time, memory) is needed to judge practicality at scale.

3. Single retrieval per prefix — diversity tradeoff.
The authors acknowledge FoSS does only one retrieval per prefix (vs. token-level retrieval in kNN-LM), which reduces latency but may limit diversity; while FoSS improves MAUVE/diversity over many baselines, kNN-LM sometimes shows higher diversity under nucleus sampling. The paper states this as a limitation but does not explore hybrid retrieval-frequency regimes.

4. Ablation could be deeper.
Ablations remove DAG (only tokens), LM, or PM, but the paper doesn’t show results for: (a) different max span lengths, (b) varying the retrieval k, (c) different segmentation probability schedules P (the stochastic early-stop thresholds), or (d) alternatives to the uniform backward policy (they chose uniform suffix PB). These choices could materially affect performance and stability.

**Questions:**

1. Segmentation sensitivity: How sensitive are results to the choice of span length bounds (2–8 tokens) and the probability schedule P used in DAG-inducing segmentation? Can you provide an ablation or plot showing MAUVE / Diversity vs. max span length and vs. different stochastic early-stop rates?

2. Reward hyperparameters & robustness: How was α (tradeoff between LM and PM) selected? Do generated distributions sharply change for small α perturbations? Have you observed reward-hacking behaviors (e.g., overly short/high-likelihood degenerate outputs) and how were they mitigated? Please report any experiments on robustness of PM (overfitting to training references, ensemble PMs, or rejection sampling).

3. Computational and latency accounting: Could you provide concrete training GPU-hour numbers and inference latency breakdowns (per prefix average, top-k retrieval, span scoring) compared to each baseline (Transformer, CoG, kNN-LM)? Table 5 is mentioned but not visible in my copy — please expand on it.

---

> ### Author Response · Authors · 2025-11-21
>
> Thank you for the detailed and constructive review. We address each point below.
>
> > **For Weakness 1 and Question 2:**
>
> - We choose the hyperparameter $\alpha$ empirically to align the scales of $\log p_{LM}$ and $\log p_{PM}$, which yields stable optimization in practice. To quickly assess sensitivity to $\alpha$, we perform an additional study on 20% of the original training data. As shown in the table below, perturbing $\alpha$ by $\pm 20\%$ changes MAUVE by only 2.2% on average, indicating that the method is robust to this hyperparameter.
>
> |              | $0.8\alpha$ | $\alpha$ | $1.2\alpha$ |
> | ------------ | ----------- | -------- | ----------- |
> | FoSS (MAUVE) | 0.297       | 0.295    | 0.284       |
>
> - The training data for the PM is described in Appendix C.3. Specifically, we use WikiText-103 training sentences as positive samples and FoSS generations from the initial policy (sampled with multiple seeds) as negative samples. We monitor PM performance on the validation set and do not observe signs of overfitting.
> - When the reward uses only the LM term $\log p_{LM}$, we observe a clear failure mode where the model generates highly repetitive spans:
>
> | Prefix                                                       | Sentence Continuation                                        |
> | ------------------------------------------------------------ | ------------------------------------------------------------ |
> | The ASD includes a Cyber Security Operations Centre ( [UNK] ) which is responsible for protecting Defence and other Australian Government agencies against cyberwarfare attacks. The [UNK] | operates jointly with Defence Security Advisory Services Operations Operations Operations Operations Operations Operations Operations Operations |
> | State Route 958 begins at an intersection with a two-lane U.S. Route 6 east of the city of Corry. The route progresses northward | through farmland bordering Buffalo Township, where it joins Route 223 northbound and Route 223 eastbound. After a while Route 223 enters Buffalo Township Parkway junction 228 exits Buffalo Township Parkway junction 228 exits Buffalo Township Parkway junction 228 exits |
>
> - In these cases, the LM assigns high likelihood to repetitive continuations. Incorporating the PM mitigates this issue because its negative samples include initial policy generations that already contain such repetitive patterns, which leads the PM to penalize these patterns.
>
> > **For Weakness 2 and Question 3:**
>
> - Thank you for this suggestion. We have added a detailed discussion on training overhead in Appendix F of the revised manuscript. Specifically,  FoSS incurs additional computational cost compared to offline methods because of its online sampling mechanism, which is inherent to RL-style training. The most comparable baseline in terms of training paradigm is GFlowNets-FT [1]. On 8 GPUs, FoSS requires 10.5 hours per epoch versus 9.25 hours for GFlowNets-FT. Our implementation builds on the GFlowNets-FT codebase without additional engineering optimizations specific to FoSS. More efficient frameworks (e.g., verl) could further reduce this overhead. Our focus in this work is the DAG-structured state space modeling for span generation with GFlowNets, rather than accelerating RL-style training.
> - Table 5 reports the end-to-end per-prefix latency in seconds; we have clarified the units in the revised manuscript. For top-K retrieval latency, FoSS and CoG use the same DPR model and the same value of $K$, resulting in equivalent time costs. For span scoring, both methods use the same GPT-2 + BERT configuration, which leads to identical scoring costs. The measured latencies of the retrieval and scoring modules (in seconds) are:
>
> |      | top-K retrieval | span scoring |
> | ---- | --------------- | ------------ |
> | CoG  | 0.029           | 0.018        |
> | FoSS | 0.029           | 0.018        |
>
> [1]  Amortizing intractable inference in large language models, ICLR 2024
>
> > **For Weakness 3:**
>
> - To improve inference efficiency, we adopt retrieval per prefix, consistent with CoG. To study the trade-off between retrieval frequency and diversity, we also evaluate variants that perform retrieval every 8 spans and every 16 spans, with results below:
>
> |           | retrieval per 8 spans | retrieval per 16 spans | retrieval per prefix |
> | --------- | --------------------- | ---------------------- | -------------------- |
> | diversity | 92.80                 | 92.62                  | 92.48                |
> | latency   | 6.81$\times$          | 3.69$\times$           | 1$\times$            |
>
> - Notably, retrieval every 8 spans incurs 6.81× the latency of retrieval per prefix, and retrieval every 16 spans incurs 3.69× the latency, while bringing only marginal diversity gains.

---

> > ### Author Response · Authors · 2025-11-21
> >
> > > **For Weakness 4 and Question 1:**
> >
> > - For span length selection, we test maximum span lengths of 4, 6, and 8 tokens:
> >
> > | **Max Span Lengths** | **4** | **6** | **8** |
> > | -------------------- | ----- | ----- | ----- |
> > | MAUVE                | 30.61 | 31.49 | 31.65 |
> >
> > - For the retrieval hyperparameter $K$, we evaluate $K \in \\{256, 512, 1024\\}$:
> >
> > | **Retrieval K** | **256** | **512** | **1024** |
> > | --------------- | ------- | ------- | -------- |
> > | MAUVE           | 32.11   | 31.79   | 31.65    |
> >
> > - For segmentation probability schedules $P$, the expected span length decreases as the threshold $p$ increases. As mentioned in Appendix E, we sample offline data using different thresholds from the set $\mathcal{P} = \\{ p_0=0, p_1, \dots, p_n \\}$ to construct the DAG state space.
> > - For backward policy, this is a core GFlowNets design choice that we plan to explore further. Following established work [2][3][4], which shows that uniform backward policies are simple to implement and empirically effective, we adopt a uniform backward policy in FoSS.
> >
> > [2] Bayesian Structure Learning with Generative Flow Networks, UAI 2022
> >
> > [3] Generative Flow Networks for Discrete Probabilistic Modeling, PMLR 2022
> >
> > [4] Trajectory balance: Improved credit assignment in GFlowNets, NeurIPS 2022

---

### Official Review · Reviewer_WyH2 · 2025-10-30

**Soundness:** 3
**Presentation:** 3
**Contribution:** 3
**Rating:** 6
**Confidence:** 3

**Summary:**

This paper proposes a novel span-based text generation framework using GFlowNets. The vocabulary is extended to include multi-word phrases, and the decoding process is reformulated as a Directed Acyclic Graph (DAG) problem. Extensive analyses and experiments are conducted to demonstrate the effectiveness of the proposed approach.

**Strengths:**

1.The paper connects well with relevant literature. Each component of the proposed FoSS model is discussed in relation to previous studies, making the distinctions from existing work clear.

2.While the method combines ideas from GFlowNets and dynamic span vocabularies (neither originally introduced by the authors), it tackles non-trivial challenges in integrating these concepts. The authors address these challenges in a thoughtful and technically sound manner.

3.The paper provides detailed explanations of each component and implementation aspect, ensuring reproducibility and transparency.

4.The experimental section is extensive, covering in-domain, out-of-domain, and downstream evaluations that collectively validate the proposed approach.

**Weaknesses:**

1.Lack of comparison with more recent and important baselines. As mentioned by the authors in the related work, two recent and important studies on dynamic vocabularies [1][2] were not included in the main experimental comparisons.

2.For scalability, the largest model evaluated in the scalability experiments is GPT2-XL (~1.5B parameters). It would be valuable to test the approach on larger and more recent models such as LLaMA 3 or Qwen 3. If direct training is infeasible due to resource constraints, a cross-model comparison (e.g., FoSS on GPT2-XL vs. vanilla LLaMA 3/Qwen 3) could still provide useful insights.

3.Although inference-time latency is reported, the paper lacks discussion of the training overhead compared to other baselines.

[1] RETRIEVAL IS ACCURATE GENERATION, ICLR 2024

[2] Nearest Neighbor Speculative Decoding for LLMGeneration and Attribution, NeurIPS 2024

**Questions:**

N.A.

---

> ### Author Response · Authors · 2025-11-21
>
> Thank you for the careful reading and for the positive overall assessment. We address the three concerns below.
>
> > **For Weakness 1:**
>
> - For [1], we were unable to reproduce its results because the official repository does not provide the training scripts and evaluation commands. For [2], the repository lacks training details for the dense index, preventing us from replacing the En-Wiki memory used in their code. Consequently, we could not evaluate this method under our in-domain and out-of-domain settings in the main experiments. Nevertheless, we implemented the method under the feasible settings, using GPT-2 as the base model, facebook/dragon-plus-query-encoder as the retrieval model, and the released En-Wiki dataset as the memory. Its MAUVE score on the Wikitext-103 test set is 0.231, which is significantly lower than FoSS.
>
> [1] RETRIEVAL IS ACCURATE GENERATION, ICLR 2024
>
> [2] Nearest Neighbor Speculative Decoding for LLM Generation and Attribution, NeurIPS 2024
>
> > **For Weakness 2:**
>
> - According to the GPT-2 paper [3], Wikitext-103 is excluded from its pretraining data. However, the pretraining corpora of recent open-source models such as LLaMA 3 or Qwen 3 are not disclosed. It is therefore unclear whether our evaluation sets are included in their pretraining corpora. This uncertainty makes direct comparisons unfair.
>
> [3] Language Models are Unsupervised Multitask Learners, OpenAI 2019
>
> > **For Weakness 3:**
>
> - Thank you for raising this point. We have added a detailed discussion on training overhead in Appendix F of the revised manuscript. Specifically,  FoSS incurs additional computational cost compared to offline methods because of its online sampling mechanism, which is inherent to RL-style training. The most comparable baseline in terms of training paradigm is GFlowNets-FT [4]. On 8 GPUs, FoSS requires 10.5 hours per epoch versus 9.25 hours for GFlowNets-FT. Our implementation builds on the GFlowNets-FT codebase without additional engineering optimizations specific to FoSS. More efficient frameworks (e.g., verl) could further reduce this overhead. Our focus in this work is the DAG-structured state space modeling for span generation with GFlowNets, rather than accelerating RL-style training.
>
> [4] Amortizing intractable inference in large language models, ICLR 2024

---

### Official Review · Reviewer_4EKG · 2025-10-31

**Soundness:** 4
**Presentation:** 4
**Contribution:** 4
**Rating:** 8
**Confidence:** 2

**Summary:**

The paper describes an approach to generating text, Flow of SpanS (FoSS), that involves using multi-token spans of existing text.  This situates it with some other recent work that also uses multi-token spans like e.g. CoG retrieving and assembling contextually relevant phrases; it differs in that it recognizes that this corresponds to a DAG-structured internal state space, which then permits the application of GFlowNets as a way of working with this structure.  This also situates it with respect to other work that uses GFlowNets, but this other work typically is applied just to single token-level generation.  Evaluations are carried out with respect to several appropriate text generation baselines under the MAUVE and Diversity metrics, showing strong performance by FoSS.  There are also evaluations on downstream tasks and ablation studies, as well as explorations of scaling.

**Strengths:**

* Overall, I think this is a great paper.  The idea is a very neat one, the sort that comes from recognizing the structure of data and realizing what this implies for methods to apply.

* The experiments are quite comprehensive, and show the effectiveness of FoSS, with e.g. quite large improvements in MAUVE scores (Table 1) and strong preferences for FoSS text (Table 2).  Beyond the values of the metrics, the case study in Fig 5 in the appendix gives a feel for how much better the text generated by FoSS can be.  I could see this approach being widely used.  (The latency analysis of App F is also encouraging as to the practicality of using the method.)

* There are several insights throughout the paper as well, as for example in the ablation discussion on the role of the PM component in Eqn (3).

**Weaknesses:**

Nothing at all major.

* There are a few places where some additional remarks on setup could help.  For instance, there’s no discussion of greedy vs nucleus in terms of setup, what they might tell us, etc; the numbers are just presented in Table 1.  Table captions in general could be a bit more informative.

**Questions:**

No questions.

---

> ### Author Response · Authors · 2025-11-21
>
> Thank you for highlighting the strengths of our work. We aim to address your concerns in this response.
>
> > **For Weakness 1:**
>
> - In the revised manuscript, we have added a clear explanation of the greedy and nucleus sampling setups in Section 4. We also expanded all table captions in Section 4 to include concise overviews of the experimental settings for each evaluation.

---

### Official Review · Reviewer_JUcH · 2025-11-01

**Soundness:** 3
**Presentation:** 3
**Contribution:** 3
**Rating:** 6
**Confidence:** 2

**Summary:**

The paper introduces FoSS, a span-based language modeling framework that builds an explicit DAG over segmentations and learns a GFlowNet policy to traverse it. Concretely, span actions (length 2–8 tokens) are drawn from a dynamic, retrieval-augmented vocabulary; training uses subtrajectory balance with a hybrid online/offline regimen, a uniform backward policy over suffixes, and a composite reward combining a fine-tuned LM likelihood and a preference model. On open-ended generation (WikiText-103), FoSS improves MAUVE over strong baselines, is preferred by GPT-4 on multiple criteria, and shows gains on several knowledge-intensive multiple-choice tasks.

**Strengths:**

- Casting span generation as an explicit DAG and optimizing it with a GFlowNet is a clean way to expose multiple compositional paths, addressing a known bias of tree-structured token decoders. I find the core idea both interesting and promising.

- The experiments are comprehensive and the results are highly positive. FoSS improves MAUVE both in-domain and out-of-domain (Table 1), shows positive GPT-4 preferences (Table 2), and is competitive on QA tasks (Table 3). The ablation study showing a drop in performance without the DAG structure supports the central claim that “many-paths-per-state” matters (Table 4).

**Weaknesses:**

- Section 4.3 (“Scaling behavior”) is somewhat misleading. In Figures 2 and 3, the x-axis is rendered with equal spacing while the labels are logarithmically spaced (0.001 → 1.0 with tripling and 0.47 → 15 with doubling). Please use a logarithmic x-axis and plot against numeric proportions to avoid misrepresenting the trends. Figure 4 should use the number of parameters as the x-axis instead of model names. The sizes for GPT-2 Small, Medium, Large, and XL are 124M, 355M, 774M, and 1.5B, respectively. Once corrected, the diminishing returns with model scaling appear quite strong, so I am not sure this method would truly generalize to larger models.
- The En-Wiki datastore (used for WikiText-103 evaluation) likely contains near-duplicates of test articles. Even if baselines share the same memory, absolute improvements can be inflated. Please filter out any pages overlapping with the WikiText-103 test set, or at least check and report overlap statistics.
- Although you cite a paper claiming that strong LLM-based evaluations mostly align with human judgments, other work suggests this alignment can be dataset-dependent (https://aclanthology.org/2025.acl-short.20/). The reliability of LLM-based evaluation remains debatable. A small human study (e.g., 100 pairs) would strengthen the conclusion.

**Questions:**

* How many training tokens and GPU-hours are used for FoSS vs. each baseline?
* Why do spans have lengths of 2–8 tokens? What motivated this design choice?

---

> ### Author Response · Authors · 2025-11-21
>
> We thank the reviewer for the detailed and constructive feedback. We address each comment and question below and have updated the manuscript accordingly.
>
> > **For Weakness 1:**
>
> - We have updated Figures 2 and 3 to use logarithmic x-axes with numeric proportions in the revised manuscript. For Figure 4, we now plot the number of parameters on a logarithmic x-axis, following the standard scaling-law visualization in [1] (Figure 1). For completeness, we also include a linear-scale version of this plot as Figure 6. The revised visualization shows that, although the absolute MAUVE gains decrease as model size increases, FoSS consistently outperforms the Transformer baseline at all scales, indicating that the method remains beneficial as model capacity grows.
>
> [1] Scaling Laws for Neural Language Models, arXiv 2020
>
> > **For Weakness 2:**
>
> - We thank the reviewer for pointing this out. We have verified that only 8 retrieved documents overlap with the WikiText-103 test set, accounting for $3.6\times 10^{-7}$ of the segmented En-Wiki corpus and 0.18% of the WikiText-103 test set. These very small ratios suggest that the impact on evaluation is minimal. We have included these overlap statistics in Appendix B.1 of the revised manuscript.
>
> > **For Weakness 3:**
>
> - Given the limited duration of the discussion period and the time required to run a human evaluation with sufficient scale and quality control, we were not able to conduct a new human study during rebuttal. We will try to include a human evaluation study in the camera-ready version. To strengthen the reliability, we additionally evaluated the generated text with two strong general-purpose models, GPT-5 and Claude Sonnet 4.5:
>
> | Model             | Better | Neutral | Worse |
> | ----------------- | ------ | ------- | ----- |
> | GPT-5             | 52%    | 28%     | 20%   |
> | Claude Sonnet 4.5 | 49%    | 17%     | 34%   |
>
> - Compared to CoG, FoSS receives more 'Better' than 'Worse' ratings from both evaluators, which is consistent with the GPT-4 based evaluation trends reported in the paper.
>
> > **For Question 1:**
>
> - FoSS uses 118,756,882 training tokens. FoSS incurs additional computational cost compared to offline methods because of its online sampling mechanism, which is inherent to RL-style training. The most comparable baseline in terms of training paradigm is GFlowNets-FT [2]. On 8 GPUs, FoSS requires 10.5 hours per epoch versus 9.25 hours for GFlowNets-FT. Our implementation builds on the GFlowNets-FT codebase without additional engineering optimizations specific to FoSS. More efficient frameworks (e.g., verl) could further reduce this overhead. Our focus in this work is the DAG-structured state space modeling for span generation with GFlowNets, rather than accelerating RL-style training.
>
> [2] Amortizing intractable inference in large language models, ICLR 2024
>
> > **For Question 2:**
>
> - We set span lengths to 2–8 tokens primarily for computational efficiency and to keep the action space manageable, and we adopt the same range as CoG for a fair comparison.

---

### Meta-Review · Area_Chair_vyNk · 2025-12-28

**Summary:**

The paper introduces FoSS, a span-based language modeling framework that builds an explicit DAG over segmentations and learns a GFlowNet policy to traverse it. Concretely, span actions (length 2–8 tokens) are drawn from a dynamic, retrieval-augmented vocabulary; training uses subtrajectory balance with a hybrid online/offline regimen, a uniform backward policy over suffixes, and a composite reward combining a fine-tuned LM likelihood and a preference model. On open-ended generation (WikiText-103), FoSS improves MAUVE over strong baselines, is preferred by GPT-4 on multiple criteria, and shows gains on several knowledge-intensive multiple-choice tasks.

**Reviewer Concerns:**

Multiple reviewers questioned whether FoSS would scale to substantially larger and newer models (e.g., LLaMA-3, Qwen-3). The authors declined such experiments due to fairness and data-leakage concerns, but this leaves the central scalability claim empirically unresolved.

While the authors corrected scaling plots, the reviewer’s concern that MAUVE gains flatten sharply at larger model sizes is not analytically addressed (e.g., no discussion of asymptotic behavior, break-even points, or regimes where FoSS may cease to be beneficial).


Despite adding GPT-5 and Claude Sonnet 4.5 evaluations, no human evaluation was conducted. The dataset-dependence concern about LLM evaluators raised by the reviewer therefore remains unresolved.

**Reviewer Scores:**

remain unchanged

---

### Decision · Program_Chairs · 2026-01-26

Accept (Poster)